# A 15-year follow-up study of hip bone mineral density and associations with leisure time physical activity. The Tromsø Study 2001–2016

**Saija Mikkilä**[1,2]*, **Jonas Johansson**[2], **Anna Nordström**[1,3], **Peter Nordström**[4], **Nina Emaus**[5], **Bjørn Helge Handegård**[6], **Bente Morseth**[1], **Boye Welde**[1]

1 School of Sport Sciences, UiT The Arctic University of Norway, Tromsø, Norway, 2 Department of Community Medicine, UiT The Arctic University of Norway, Tromsø, Norway, 3 Department of Public Health and Clinical Medicine, Section for Sustainable Health, Umeå University, Umeå, Sweden, 4 Department of Community Medicine and Rehabilitation, Geriatric Medicine, Umeå University, Umeå, Sweden, 5 Department of Health and Care Sciences, UiT The Arctic University of Norway, Tromsø, Norway, 6 The Regional Centre for Child and Adolescent Mental Health–North, UiT The Arctic University of Norway, Tromsø, Norway

* saija.p.mikkila@uit.no

**Data Availability Statement:** The legal restriction on data availability are set by the Tromsø Study Data and Publication Committee in order to control

## Abstract

### Aims

The aim was to investigate the long-term association between leisure time physical activity and hip areal bone mineral density (aBMD), in addition to change in hip aBMD over time, in 32–86 years old women and men.

### Methods

Data were retrieved from the 2001, 2007–2008, and 2015–2016 surveys of the Tromsø Study, a longitudinal population study in Norway. Leisure time physical activity was assessed by the four-level Saltin-Grimby Physical Activity Level Scale which refers to physical exertion in the past twelve months. Hip aBMD was assessed by Dual-Energy X-ray Absorptiometry. Linear Mixed Model analysis was used to examine long-term associations between physical activity and hip aBMD (n = 6324). In addition, the annual change in hip aBMD was analyzed in a subsample of 3199 participants.

### Results

Physical activity was significantly and positively associated with total hip aBMD in the overall cohort (p<0.005). Participants who reported vigorous activity had 28.20 mg/cm$^2$ higher aBMD than those who were inactive (95% CI 14.71; 41.69, controlled for confounders), and even light physical activity was associated with higher aBMD than inactivity (8.32 mg/cm$^2$, 95% CI 4.89; 11.76). Associations between physical activity and femoral neck aBMD yielded similar results. Hip aBMD decreased with age in both sexes, although more prominently in women. From 2001 to 2007–2008, aBMD changed by –5.76 mg/cm$^2$ per year (95% CI –6.08; –5.44) in women, and –2.31 mg/cm$^2$ (95% CI –2.69; –1.93), in men. From 2007–2008 to 2015–2016, the change was –4.45 mg/cm$^2$ per year (95% CI –4.84; –4.06) in women, and –1.45 mg/cm$^2$ (95% CI –1.92; –0.98) in men.

for data sharing, including publication of datasets with the potential of reverse identification of de-identified sensitive participant information. The data can however be made available from the Tromsø Study upon application to the Tromsø Study Data and Publication Committee. Contact information: The Tromsø Study, Department of Community Medicine, Faculty of Health Sciences, UiT The Arctic University of Norway; e-mail: tromsous@uit.no.

**Funding:** SM's work was funded by Norwegian Women's Public Health Association (https://sanitetskvinnene.no/english). The publication charges for this article have been funded by a grant from the publication fund of UiT The Arctic University of Norway. The funders had no role in study design, data collection and analysis, decision to publish, or preparation of the manuscript.

**Competing interests:** The authors have declared that no competing interests exist.

## Conclusions

In this cohort of adult men and women, physical activity levels were positively associated with hip aBMD in a dose-response relationship. Hip aBMD decreased with age, although more pronounced in women than men.

## Introduction

Osteoporosis is characterized by reduced bone mineral density (BMD) and poses a worldwide health threat to ageing populations, mainly by increasing the risk of low impact fractures [1]. BMD is a strong predictor of hip fractures, considered to be the most severe among fragility fractures, as they lead to higher morbidity and 14–58% increased mortality risk one year after a hip fracture [2–4]. Furthermore, hip fractures contribute to extensive economic costs in form of hospitalization and rehabilitation [3]. Still, the recovery rate is relatively low compared to other types of fractures [3], and other detrimental health conditions such as walking disabilities are common among hip fracture patients [5,6].

Numerous studies indicate that physical activity improves BMD and thereby reduces the risk of hip fractures [7–9]. However, previous cohort studies on physical activity and BMD have used methodological approaches characterized by small sample sizes, cross-sectional designs, or subpopulations [10–14]. Long-term effects of physical activity on BMD have previously been shown among adults and older adults separately [15–17], although this relationship should be further elucidated across a broad age spectrum in the general population.

The evidence indicating benefits of physical activity on hip BMD could potentially be strengthened by involving large cohort studies, and advanced statistical approaches utilizing longitudinal data. Therefore, the aim of this prospective study was to investigate the association between leisure time physical activity and hip BMD with up to 15 years follow-up in women and men who participated in the population-based Tromsø Study surveys in 2001 (Tromsø5), 2007–2008 (Tromsø6) and 2015–2016 (Tromsø7). In addition, we wanted to examine potential changes in hip BMD with ageing in a large cohort of middle-aged and older adults.

## Methods

### Design, subjects, and ethical approval

The Tromsø Study is a population-based study [18] with repeated surveys conducted in the municipality of Tromsø, Norway. The study was initiated in 1974, and continued with six follow-up surveys in 1979–1980, 1986–1987, 1994–1995, 2001, 2007–2008 and in 2015–2016. All seven surveys include Tromsø municipality residents who were invited to participate through invitation letters. The survey cohorts included total population, birth cohorts and/or random samples of inhabitants in the Tromsø municipality. All participants provided written informed consent prior to inclusion. The Tromsø Study has been approved by the Data Inspectorate of Norway and the Regional Committee of Medical and Health Research Ethics, North Norway (2009/2536, 2010/876 and 2014/940).

This study was based on the Tromsø5, Tromsø6 and Tromsø7 surveys of the Tromsø Study including a total of 7790 participants; 4468 women and 3322 men, aged 32–86 years by the first dual-energy X-ray absorptiometry (DXA) scan. Participants with at least one valid hip areal BMD (aBMD) measurement were included in the prospective follow-up analysis. We

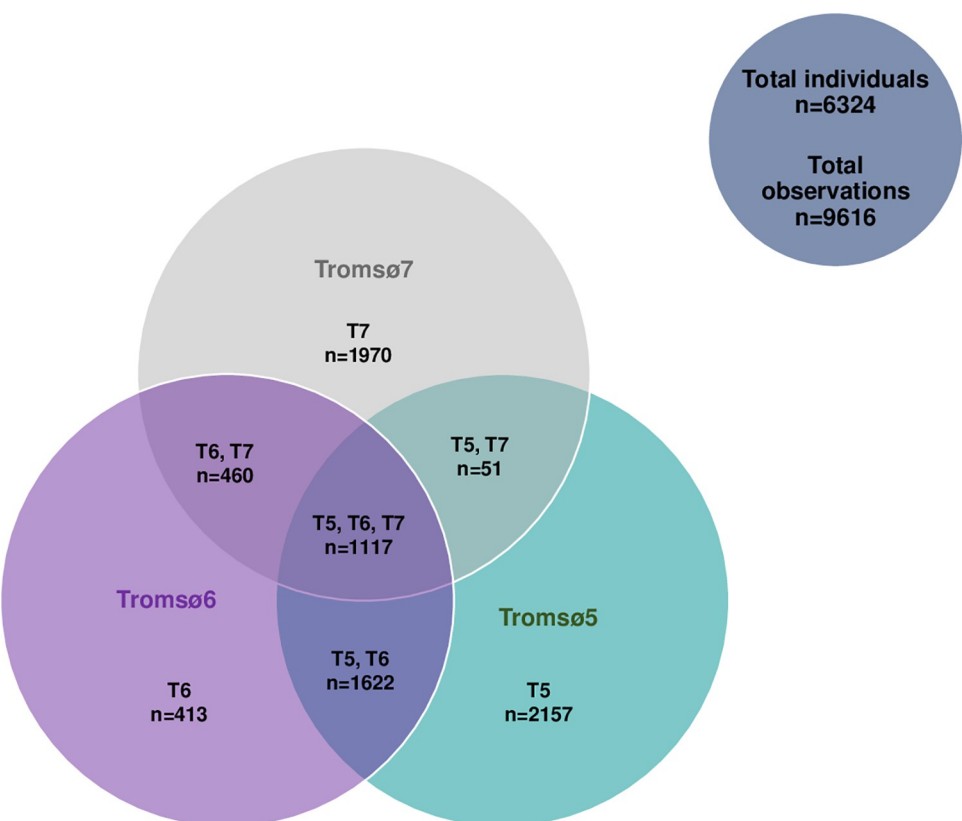

**Fig 1. Number and distribution of the participants in the Tromsø Study surveys Tromsø5, Tromsø6 and Tromsø7.**

investigated associations between physical activity and left hip aBMD by using the Tromsø Study data from Tromsø5, Tromsø6 and Tromsø7 in 6324 participants; 3637 women and 2687 men, aged 32–86 years. A total of 9616 observations were included, as each participant could contribute with data on any of the three measuring points; Tromsø5, Tromsø6 and/or Tromsø7. Furthermore, we investigated change in hip BMD in 3199 participants; 1938 women and 1612 men, aged 40–84 years. The total number of scans for the change in the left hip total aBMD was 3947 as participants with at least two consecutive hip aBMD measurements, Tromsø5 and Tromsø6 or/and Tromsø6 and Tromsø7, were included. Fig 1 shows the number of participants and their Tromsø Study survey distribution.

## Assessment of physical activity

Leisure time physical activity was assessed by the Saltin-Grimby Physical Activity Level Scale (SGPALS) [19], which is a self-administrated multiple-choice question included in a questionnaire on several lifestyle and health-related questions. Questionnaires for the Tromsø5, Tromsø6 and Tromsø7 surveys concerning leisure time exercise and physical exertion accompanied the study invitation. The SGPALS asks about "Exercise and physical exertion in leisure time. If your activity varies much, for example between summer and winter, then give an average. The question refers only to the last twelve months". The participants were asked to choose one of the following response options: 1) Reading, watching TV, or other sedentary activity; 2) Walking, cycling, or other forms of exercise at least 4 hours a week (Including walking or cycling to place of work, Sunday-walking, etc.); 3) Participation in recreational sports, heavy

gardening, etc. (Note: duration of activity at least 4 hours a week); 4) Participation in hard training or sports competitions, regularly several times a week. The participants were according to their response option allocated into one of the four levels of physical activity [19].

## Measurement of bone mineral density

In the Tromsø5, Tromsø6 and Tromsø7 surveys, aBMD was measured by using DXA devices (Lunar Prodigy, GE Medical Systems, Madison, WI, USA) in a randomly selected sub-sample. The first Lunar Prodigy Pro device was used for the Tromsø5 and Tromsø6 measurements, and the second, cross-calibrated with the first device, for Tromsø7. Valid measurements were obtained from 7790 individuals. All scans were performed according to standard procedures set by GE Medical Systems. The DXA device was calibrated daily throughout the surveys using a standard phantom. Trained technicians performed the scanning according to the standardized protocol, and one of them performed quality assessment of the total sample afterwards. In a validation study, the short-term in vivo precision error for the Lunar Prodigy was 1.7% and 1.2% for the femoral neck and total hip measurements, respectively [20]. Our main analyses are based on left total hip scans, which include the femoral neck, trochanter and shaft regions [21]. In a sensitivity analysis we included left femoral neck instead of total hip.

## Additional measurements

Participants' height and weight were measured at the physical examination in light clothing to nearest centimeter and half-kilogram respectively. Body mass index (BMI) was calculated from weight and height ($kg/m^2$). Smoking (current, previous or never) was self-reported.

## Statistical analyses

Descriptive data are presented as means (M) and standard deviations (SD), or as number of participants (N) and percentages (%). Participants with at least two consecutive hip aBMD measurements, Tromsø5 and Tromsø6 or/and Tromsø6 and Tromsø7, were included in the hip aBMD change analysis. Change per year in the left total hip aBMD was studied from Tromsø5 to Tromsø6 and from Tromsø6 to Tromsø7, stratified by 5-year age groups, separately for men and women. From Tromsø5 to Tromsø6, the highest age group 75+, also included subjects above 80 years due to low count of subjects over 75 years at the Tromsø5 baseline. From Tromsø6 to Tromsø7, the highest age group 75+, included subjects < 77 years at the Tromsø6 baseline.

Participants with at least one hip aBMD measurement and data for physical activity, age, BMI, sex and smoking status were included in LMM analysis, and associations between left total hip/femoral neck aBMD and physical activity, adjusted for age, BMI, sex and smoking status were analyzed with an LMM approach. This approach handles the dependency in aBMD observations within individuals, making this a 2-level analysis with observations nested within persons [22]. LMM is analyzed with data in a long format, and therefore handles time-varying independent variables like physical activity. In the LMM, a random intercept and a random slope for age were included in the model. Also, answers for the leisure time physical activity questions were recoded for the linear mixed model analysis: 1->4, 2->3, 3->2, 4->1, and the level 4 (inactive) functioned as a reference group.

We tested interaction terms by involving physical activity first in order to be able to conclude whether the main effect of physical activity level (PA) on the left total hip aBMD should be interpreted. No significant effects of either PA*sex, PA*BMI, PA*age or PA*smoking were found. We included a sex*age interaction term in all models tested.

Since we have a large sample, an alpha level of 0.005 was chosen for testing the whole sample [23]. 95% confidence intervals were computed when estimating parameters within subgroups of the sample. SPSS version 26 (IBM Corp, Armonk, NY, USA) was used for all analyses.

## Results

### Sample characteristics

Table 1 displays sample characteristics stratified by the participants' first Tromsø Study survey examination, for women and men separately. The table presents all 7790 participants with at least one aBMD measurement. Women had a mean age of 63.1 ± 9.2 years and a BMI of 26.7 ± 4.6 kg/m$^2$, while corresponding values for men were 63.9 ± 9.2 years and 27.1 ± 3.6 kg/m$^2$.

### Physical activity and left hip aBMD

Table 2 shows long-term associations between leisure time physical activity and total hip aBMD, adjusted for sex, age, sex*age, BMI and smoking habits, in women and men who participated in the population-based Tromsø Study surveys in Tromsø5, Tromsø6 and Tromsø7. Physical activity was significantly and positively associated with left total hip aBMD in the overall cohort ($F_{3, 3424.8}$ = 14.38; $p < 0.005$). Compared with the most inactive participants, aBMD increased gradually with increasing physical activity level, and participants who reported vigorous activity had 28.20mg/cm$^2$ higher aBMD than those who were inactive ($t_{3666.9}$ = 4.10; $p < 0.005$) (95% CI 14.71; 41.69). Associations between physical activity and femoral neck aBMD were very similar, and the correlation coefficient between total hip aBMD and femoral neck aBMD was approximately 0.90 on all measurement occasions in T5-T7.

### Changes in left total hip aBMD

Changes in aBMD from Tromsø5 to Tromsø6 (women: $n$ = 1419; men: $n$ = 992) stratified by 5-year age groups are shown in Table 3. From Tromsø5 to Tromsø6, the change per year in aBMD was smallest at the age of 40–44 years (–2.84 mg/cm$^2$) in women. From the age of 45, the magnitude of the annual change was markedly larger, ranging from –4.87 to –7.81 mg/cm$^2$. In men, aBMD changed progressively and significantly from 55–59 years and up, with a markedly higher decrease in the oldest ($\geq$ 75 years) group compared to younger participants. Overall, the annual aBMD change was –5.76 mg/cm$^2$ (95% CI –6.08; –5.44) in women, and –2.31 mg/cm$^2$ (95% CI –2.69; –1.93) in men.

Changes in aBMD from Tromsø6 to Tromsø7 (women: $n$ = 907; men: $n$ = 629) stratified by 5-year age group are shown in Table 4. In women, the patterns in aBMD decrease were similar to the patterns observed between Tromsø5 and Tromsø6. In men, the aBMD started decreasing significantly from 65–69 years, as opposed to ages 55–59 from Tromsø5 to Tromsø6. The aBMD loss was markedly higher in the oldest ($\geq$ 75 years) age groups (–6.41 mg/cm$^2$). Mean annual aBMD change was –4.45 mg/cm$^2$ (95% CI –4.84; –4.06) in women, and –1.45 mg/cm$^2$ (95% CI –1.92; –0.98) in men.

## Discussion

In this 15-year longitudinal study of adult and elderly women and men, physical activity was positively and linearly associated with hip aBMD. Moreover, the annual decrease in aBMD was rather stable from the age of 45 years in women and from the age of 55–65 years in men.

**Table 1. Sample characteristics stratified by sex and survey participation.**

| Women | T5 | T6 | T7 | T6, T7 | T5, T6 | T5, T7 | T5, T6, T7 | Total |
|---|---|---|---|---|---|---|---|---|
| **Age** (years) | | | | | | | | |
| n | 1221 | 200 | 1079 | 259 | 967 | 30 | 712 | 4468 |
| M (SD) | 67.9 (9.5) | 59.3 (7.3) | 62.0 (7.9) | 57.1 (5.1) | 64.7 (8.5) | 55.1 (6.7) | 58.2 (8.0) | 63.1 (9.2) |
| **BMI** (kg/m$^2$) | | | | | | | | |
| n | 1216 | 200 | 1078 | 259 | 962 | 30 | 710 | 4455 |
| M (SD) | 26.9 (4.7) | 26.4 (4.8) | 26.7 (4.8) | 26.3 (4.2) | 27.1 (4.5) | 25.4 (3.9) | 26.4 (4.2) | 26.7 (4.6) |
| **Smoking** n (%) | 1209 (27.3) | 199 (4.5) | 1063 (24.0) | 255 (5.8) | 30 (0.7) | 961 (21.7) | 706 (16.0) | 4423 (100.0) |
| Yes, now | 348 (28.8) | 58 (29.1) | 139 (13.1) | 44 (17.3) | 7 (23.3) | 237 (24.7) | 176 (24.9) | 1009 (22.8) |
| Never | 532 (44.0) | 59 (29.6) | 412 (38.8) | 81 (31.8) | 11 (36.7) | 432 (45.0) | 294 (41.6) | 1821 (41.2) |
| Yes, previously | 329 (27.2) | 82 (41.2) | 512 (48.2) | 130 (51.0) | 12 (40.0) | 292 (30.4) | 236 (33.4) | 1593 (36.0) |
| **Hip aBMD** (mg/cm$^2$) | | | | | | | | |
| n | 1221 | 200 | 1079 | 259 | 967 | 30 | 712 | 4468 |
| M (SD) | 869 (142) | 921 (145) | 928 (136) | 940 (115) | 911 (136) | 920 (109) | 946 (125) | 911 (138) |
| **Physical activity*** n (%) | 556 (16.8) | 179 (5.4) | 1046 (31.6) | 240 (7.2) | 607 (18.3) | 28 (0.8) | 656 (19.8) | 3312 (100.0) |
| Inactive (Level 1) | 105 (18.9) | 38 (21.2) | 103 (9.8) | 40 (16.7) | 103 (17.0) | 4 (14.3) | 100 (15.2) | 493 (14.9) |
| Light activity (Level 2) | 392 (70.5) | 119 (66.5) | 743 (71.0) | 161 (67.1) | 441 (72.7) | 22 (78.6) | 492 75.0) | 2370 (71.6) |
| Moderate activity (Level 3) | 56 (10.1) | 21 (11.7) | 184 (17.6) | 35 (14.6) | 59 (9.7) | 2 (7.1) | 62 (9.5) | 419 (12.7) |
| Vigorous activity (Level 4) | 3 (0.5) | 1 (0.6) | 16 (1.5) | 4 (1.7) | 4 (0.7) | 0 (0.0) | 2 (0.3) | 30 (0.9) |
| **Men** | **T5** | **T6** | **T7** | **T6, T7** | **T5, T6** | **T5, T7** | **T5, T6, T7** | **Total** |
| **Age** (years) | | | | | | | | |
| n | 936 | 213 | 891 | 201 | 655 | 21 | 405 | 3322 |
| M (SD) | 68.8 (9.0) | 58.8 (7.0) | 62.1 (8.4) | 58.8 (5.6) | 65.7 (8.6) | 55.4 (7.9) | 59.0 (8.1) | 63.9 (9.2) |
| **BMI** (kg/m$^2$) | | | | | | | | |
| n | 928 | 213 | 891 | 201 | 654 | 21 | 404 | 3312 |
| M (SD) | 26.5 (3.5) | 27.5 (3.9) | 27.7 (4.0) | 27.8 (3.7) | 27.0 (3.2) | 26.9 (1.8) | 26.9 (3.0) | 27.1 (3.6) |
| **Smoking** n (%) | 932 (28.6) | 209 (6.4) | 842 (25.8) | 199 (6.1) | 652 (20.0) | 21 (0.6) | 403 (12.4) | 3258 (100.0) |
| Yes, now | 272 (29.2) | 49 (23.4) | 91 (10.8) | 21 (10.6) | 154 (23.6) | 2 (9.5) | 88 (21.8) | 677 (20.8) |
| Never | 158 (17.0) | 65 (31.1) | 306 (36.3) | 74 (37.2) | 127 (19.5) | 10 (47.6) | 119 (29.5) | 859 (26.4) |
| Yes, previously | 502 (53.9) | 95 (45.5) | 445 (52.9) | 104 (52.3) | 371 (56.9) | 9 (42.9) | 196 (48.6) | 1722 (52.9) |
| **Hip aBMD** (mg/cm$^2$) | | | | | | | | |
| n | 936 | 213 | 891 | 201 | 655 | 21 | 405 | 3322 |
| M (SD) | 997 (148) | 1027 (142) | 1061 (155) | 1035 (143) | 1022 (134) | 1089 (94) | 1045 (125) | 1030 (146) |
| **Physical activity*** n (%) | 417 (16.8) | 195 (7.9) | 871 (35.1) | 196 (7.9) | 400 (16.1) | 21 (0.8) | 379 (15.3) | 2479 (100.0) |
| Inactive (Level 1) | 88 (21.1) | 51 (26.2) | 113 (13.0) | 34 (17.3) | 76 (19.0) | 3 (14.3) | 61 (16.1) | 426 (17.2) |
| Light activity (Level 2) | 256 (61.4) | 103 (52.8) | 480 (55.1) | 110 (56.1) | 256 (64.0) | 13 (61.9) | 241 (63.3) | 1459 (58.9) |
| Moderate activity (Level 3) | 69 (16.5) | 38 (19.5) | 260 (29.9) | 50 (25.5) | 63 (15.8) | 5 (23.8) | 72 (19.0) | 557 (22.5) |
| Vigorous activity (Level 4) | 4 (1.0) | 3 (1.5) | 18 (2.1) | 2 (1.0) | 5 (1.3) | 0 (0.0) | 5 (1.3) | 37 (1.5) |

*Physical activity level 1 = answer alternative 1 (lowest physical activity level); level 4 = answer alternative 4 (highest physical activity level). Survey participation is highlighted in bold dark font, and characteristics are presented at baseline if several surveys were attended. T5: the fifth Tromsø Study in 2001, T6: the sixth Tromsø Study in 2007–2008, T7: the seventh Tromsø Study in 2015–2016.

Physical inactivity is known to be an important risk factor for bone health [7], and positive associations between physical activity and hip BMD in different populations are well documented in cross-sectional studies [10,13,24–26], although such study designs are vulnerable to reverse causation. Similar findings have also been reported from randomized controlled trials [27–29], however these involve different inclusion criteria, and typically investigate specific

**Table 2. Association between leisure time physical activity, confounders, and the left hip total aBMD (mg/cm$^2$).**

| | B | 95% CI | F/ df |
|---|---|---|---|
| **Physical activity** | | | 14.38*/ 3, 3424.8 |
| Inactive (Level 1) | Reference | Reference | |
| Light activity (Level 2) | 8.32 | (4.89, 11.76) | |
| Moderate activity (Level 3) | 13.56 | (9.07, 18.05) | |
| Vigorous activity (Level 4) | 28.20 | (14.71, 41.69) | |
| **Age** (per year, slope for men) | −1.93 | (−0.23, −1.63) | 1317.8**/ 1, 2495.2 |
| **BMI** (per kg/m$^2$) | 10.54 | (9.95, 11.12) | 1244.3**/ 1, 7894.5 |
| **Sex[a]** | | | 1352.4**/ 1, 5557.3 |
| Men | Reference | Reference | |
| Women | −124.26 | (−130.88, −117.63) | |
| **Sex*Age[b]** | | | 323.9**/ 1, 2367.1 |
| Men*age | Reference | Reference | |
| Women*age | −3.62 | (−4.01, −3.22) | |
| **Smoking** | | | 17.8*/ 2, 5508.9 |
| Never | Reference | Reference | |
| Yes, previously | −16.07 | (−21.38, −10.77) | |
| Yes, now | −15.26 | (−21.52, −9.01) | |

[a] computed at the mean age of 66.1 years.

[b] difference in age slope for women and men.

* p < .005.

** p < 0.0005.

forms of physical activity, such as resistance exercise and high-impact training, which might not be generalizable to the entire population. It is therefore important to further clarify the relationship between habitual physical activity and bone health in the general population using longitudinal study designs.

This study expands a 22-year follow-up study of associations between leisure time physical activity and hip BMD in earlier Tromsø Study cohorts (1979–1980 and 2001–2002) of women and men aged 20–54 years at baseline [17]. Although the subjects were younger at baseline and follow-up, their findings were similar to our study, thus indicating positive associations

**Table 3. Change per year in age-stratified left total hip bone mineral density (mg/cm$^2$ and percent) from Tromsø5 to Tromsø6.**

| | Women | | | | | Men | | | | |
|---|---|---|---|---|---|---|---|---|---|---|
| Age group | N | Mean | SD | 95% CI | Annual change (%) | N | Mean | SD | 95% CI | Annual change (%) |
| 40–44 | 46 | −2.84 | 7.71 | (−4.60, −1.07) | −0.30 | 18 | −0.72 | 6.30 | (−3.54, 2.11) | −0.04 |
| 45–49 | 41 | −7.41 | 7.82 | (−9.28, −5.54) | −0.75 | 30 | 0.19 | 3.99 | (−2.00, 2.37) | 0.02 |
| 50–54 | 54 | −7.81 | 7.25 | (−9.43, −6.18) | −0.80 | 76 | −0.71 | 4.42 | (−2.08, 0.66) | −0.07 |
| 55–59 | 368 | −6.01 | 7.14 | (−6.63, −5.38) | −0.62 | 122 | −1.62 | 4.88 | (−2.70, −0.53) | −0.17 |
| 60–64 | 413 | −5.75 | 6.24 | (−6.34, −5.16) | −0.61 | 263 | −2.16 | 5.50 | (−2.90, −1.42) | −0.22 |
| 65–69 | 249 | −4.87 | 6.27 | (−5.62, −4.11) | −0.53 | 250 | −2.59 | 5.21 | (−3.35, −1.84) | −0.26 |
| 70–74 | 163 | −6.04 | 5.55 | (−6.98, −5.10) | −0.70 | 158 | −2.72 | 5.38 | (−3.67, −1.77) | −0.29 |
| 75+* | 85 | −6.35 | 6.78 | (−7.65, −5.05) | −0.75 | 75 | −5.16 | 6.74 | (−6.54, −3.77) | −0.53 |
| Total | 1419 | −5.76 | 6.62 | (−6.08, −5.44) | −0.62 | 992 | −2.31 | 5.43 | (−2.69, −1.93) | −0.24 |

* The highest age group 75+ includes subjects > 80 years. Age groups are based on participants' age at T5.

**Table 4. Change per year in age-stratified left hip total bone mineral density (mg/cm$^2$ and percent) from Tromsø6 to Tromsø7.**

| Age group | Women | | | | | Men | | | | |
|---|---|---|---|---|---|---|---|---|---|---|
| | N | Mean | SD | 95% CI | Annual change (%) | N | Mean | SD | 95% CI | Annual change (%) |
| 40–44 | 23 | −2.98 | 5.59 | (−5.39, −0.57) | −0.32 | 17 | 1.30 | 5.94 | (−1.75, 4.35) | 0.10 |
| 45–49 | 35 | −6.29 | 6.28 | (−8.24, −4.33) | −0.66 | 13 | 0.61 | 3.08 | (−2.88, 4.10) | 0.06 |
| 50–54 | 108 | −5.63 | 6.43 | (−6.74, −4.51) | −0.56 | 75 | 0.53 | 5.00 | (−0.92, 1.99) | 0.05 |
| 55–59 | 122 | −4.13 | 5.78 | (−5.18, −3.09) | −0.44 | 110 | −0.73 | 5.14 | (−1.93, 0.47) | −0.07 |
| 60–64 | 237 | −3.66 | 5.85 | (−4.41, −2.91) | −0.39 | 143 | −0.48 | 5.27 | (−1.53, 0.57) | −0.05 |
| 65–69 | 247 | −4.84 | 6.47 | (−5.57, −4.10) | −0.51 | 142 | −2.09 | 5.58 | (−3.14, −1.03) | −0.22 |
| 70–74 | 107 | −3.58 | 6.34 | (−4.70, −2.46) | −0.39 | 109 | −3.76 | 5.51 | (−4.97, −2.56) | −0.37 |
| 75+* | 28 | −6.81 | 8.00 | (−8.99, −4.63) | −0.72 | 20 | −6.41 | 6.88 | (−8.99, −3.82) | −0.65 |
| Total | 907 | −4.45 | 6.27 | (−4.84, −4.06) | −0.47 | 629 | −1.45 | 5.60 | (−1.92, −0.98) | −0.15 |

* In the highest age group 75+, all subjects are <77 years. Age groups are based on participants' age at Tromsø6.

between physical activity and BMD across the lifespan. Kemmler et al. [30] found that exercise had favorable effect on hip and lumbar spine BMD in their 16-year follow-up study of early-postmenopausal osteopenic women. Similarly, large cohort studies show that higher intensity of physical activity is associated with higher forearm BMD in premenopausal [31] and post-menopausal [16] women.

Our findings are comparable with the findings of a 27-year follow-up study in men, although comparability is hampered by different measurements sites; whole body and lumbar spine, and participants' young age (13 years) at baseline [32]. In a cross-sectional study, vigorous PA showed the strongest positive association with left femoral neck in 70-year-old men and women, whereas no association was found between PA of any intensity and aBMD of the left radius or lumbar spine [14]. Also, the vertical impacts were found to be stronger than other axial directions [14]. Vertical impact from e.g. running or jumping could be more prominently associated with hip BMD than lumbar spine BMD, or require higher intensity to gain the same effect on lumbar spine[14], which has also been confirmed in experimental studies [27–29]. Further, in a 15-year follow-up study bone loss at the hip was associated with long-term PA, whereas no associations of PA and bone loss in lumbar spine were seen in postmenopausal women [33].

Our results show that participants reporting vigorous physical activity had on average 28.2 mg/cm$^2$ higher aBMD in the left total hip compared with inactive participants, after adjusting for multiple confounders, which corresponds to roughly 3% of the sample mean. To put this into context, a recent randomized clinical trial showed a 5.1% increase in total hip BMD after 15 months of pharmaceutical treatment [34]. However, encouraging inactive individuals to become vigorously physically active is challenging, and physical activity is likely not the sole solution to improving or maintaining BMD in the population. However, the abundance of beneficial health effects that stems from becoming more physically active should not be ignored.

The results from this study should be viewed in light of the following limitations. The SGPALS does not provide specific information on impact direction or type of PA that participants engage in [19], which is also important to consider, because activities such as cycling or swimming might not have the same effects on hip BMD as e.g. running. Also, self-reported physical activity is subject to recall bias and social desirability bias. Nevertheless, in a substudy of Tromsø6, Emaus et al. [35] concluded that the SGPALS has acceptable validity against objectively measured physical activity assessed by the ActiGraph accelerometer in 313 healthy

men and women aged 40–44 years. Similarly, in a population-based cohort of 4040 men and women, Sagelv et al. [36] found positive associations between ranking of physical activity using the SGPALS and accelerometry measured physical activity (p<0.001), although the correlations between SGPALS and accelerometry estimates were weak (r = 0.11 to 0.26, p<0.001). No objective physical activity data to support the subjective measurements was accessible from the Tromsø study in 2001 and 2007–2008. Moreover, our analyses did not include potential confounders such as dietary factors, medication affecting bone metabolism or general health status. The most active participants in our study may have adopted a healthier lifestyle in general, which may have influenced our findings of higher bone mass in this segment. For example, previous studies have shown that nonsmoking and a high physical activity level, as well as a high body weight, reduces bone loss in both sexes [37]. Finally, it should be noted that we have not analyzed other BMD sites, as longitudinal data is not available for e.g. spine.

## Conclusion

In this cohort of adult and elderly women and men, physical activity was positively associated with left total hip aBMD in a dose-response relationship, after controlling for age, sex, BMI and smoking status. Furthermore, our findings suggest that the decrease in left total hip aBMD is more prominent with ageing in women than in men, although found in both sexes.

Future studies on this topic might benefit from combining objectively measured physical activity data, such as accelerometry measured physical activity, with additional information about the nature of the activity from self-reports in order to advance the knowledge in the field of PA and BMD. As physical activity is a complex behavior to measure, and accelerometers have limitations providing information on activities such as swimming, cycling, and weightlifting [38], combining methods and developing a more valid questionnaire for measuring bone specific physical activity, would be beneficial in this area of research.

## Implications

Although physical activity is positively associated with left total hip aBMD, the effect magnitude of self-reported physical activity is lower than the effect magnitude related to age, sex and BMI. The clinical significance of higher aBMD with higher physical activity levels is difficult to estimate as fracture risk depends on several factors. In general, low hip BMD is a strong predictor of hip fractures in men and women [39], and physical activity should be encouraged in order to prevent BMD loss and thereby reduce fracture risk.

## Author Contributions

**Conceptualization:** Saija Mikkilä, Nina Emaus, Bente Morseth.

**Formal analysis:** Bjørn Helge Handegård.

**Funding acquisition:** Saija Mikkilä, Bente Morseth.

**Investigation:** Nina Emaus.

**Methodology:** Saija Mikkilä, Nina Emaus, Bjørn Helge Handegård, Bente Morseth.

**Project administration:** Saija Mikkilä, Bente Morseth, Boye Welde.

**Resources:** Anna Nordström, Peter Nordström, Nina Emaus, Bente Morseth.

**Supervision:** Nina Emaus, Bente Morseth, Boye Welde.

**Validation:** Bjørn Helge Handegård.

**Visualization:** Saija Mikkilä, Jonas Johansson, Bjørn Helge Handegård, Bente Morseth, Boye Welde.

**Writing – original draft:** Saija Mikkilä, Jonas Johansson, Bjørn Helge Handegård, Bente Morseth, Boye Welde.

**Writing – review & editing:** Saija Mikkilä, Jonas Johansson, Anna Nordström, Peter Nordström, Nina Emaus, Bjørn Helge Handegård, Bente Morseth, Boye Welde.

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
