## [Decision Letter · Decision Letter 0]

6 Oct 2021

PONE-D-21-28316A 15-year follow-up study of hip bone mineral density and associations with leisure time physical activity. The Tromsø Study 2001-2016.PLOS ONE

Dear Dr. Mikkilä,

Thank you for submitting your manuscript to PLOS ONE. After careful consideration, we feel that it has merit but does not fully meet PLOS ONE’s publication criteria as it currently stands. Therefore, we invite you to submit a revised version of the manuscript that addresses the points raised during the review process.

Both reviewers suggest edits to your MS, which it would be desirable to address.  However, please note that these are suggestions, not mandates.

We look forward to receiving your revised manuscript.

Kind regards,

Robert Daniel Blank, MD, PhD

Academic Editor

PLOS ONE

Journal Requirements:

Reviewers' comments:

Reviewer's Responses to Questions

**Comments to the Author**

1. Is the manuscript technically sound, and do the data support the conclusions?

Reviewer #1: Yes

Reviewer #2: Yes

2. Has the statistical analysis been performed appropriately and rigorously? 

Reviewer #1: Yes

Reviewer #2: Yes

3. Have the authors made all data underlying the findings in their manuscript fully available?

Reviewer #1: No

Reviewer #2: No

4. Is the manuscript presented in an intelligible fashion and written in standard English?

Reviewer #1: Yes

Reviewer #2: Yes

5. Review Comments to the Author

Reviewer #1: Well written manuscript. Minor comment

The sample is probably a relatively healthier population than in Australia (or the USA) judged from their BMI below (in Australia 41% of subjects in this age group are obese based on BMI https://www.aihw.gov.au/reports/australias-health/overweight-and-obesity)

Study: Women had BMI of 26.7 ± 4.6 kg/m2 Men 27.1 ± 3.6 kg/m2

It is likely that the segment of this relatively non obese population that is most active is also the healthiest and would eat better, probably drink less, and possibly smoke less (although they tried to account for this). The authors are aware of this limitation and refer to it quicky in the final paragraph before the Conclusion when they say:

"Also, our analysis did not include additional confounders such as dietary factors, medication affecting bone metabolism or general health status."

This is a bit brief and should be expanded by saying it is possible that the more physically active cohort would most likely be the 'healthiest cohort' which is an alternative explanation for the findings.

Reviewer #2: This is an interesting, rigorously conducted and well-reported study showing that higher physical activity levels are associated with greater hip aBMD in a large cohort of adults and older adults. The subjective categorisation of physical activity is a limitation of the study, however the authors have thoroughly discussed this in the limitations section. Some general and specific comments are provided below.

General comments:

• There are very few participants categorised as physical activity level 4. I wonder whether combining physical activity categories 3 and 4 (as was done in the Morseth et al 2010 paper; reference 18) would have any influence on the results.

• It would be interesting to see if physical activity is similarly associated with femoral neck aBMD. Including femoral neck BMD data would strengthen the study.

• Reporting BMD results as percentage differences/changes (as opposed to absolute values in mg/cm2) would be more easily interpreted by the reader.

Specific Comments:

• Page 2, line 28: Given it is a key component of the study, I think it should be made clear in the abstract that the questionnaire categorised participants into physical activity levels (as opposed to estimated volume/intensity of physical activity). Wording such as ‘Categories of physical activity were determined by questionnaire’ would be more informative.

• The results section should be presented in the same order as the study aims. To align with the order of the aims presented at the end of the introduction, the ‘Changes in left total hip aBMD’ and ‘Physical activity and left total hip aBMD’ sections in the results should be swapped.

• Page 16, lines 296-300: It is mentioned here that physical activity was assessed by accelerometry in a small sample in 2007-08, and in a larger sample in 2015-16. The next sentence says that objective physical activity data was not available in 2007-08. Does this mean that the data used in the Emaus et al. paper cited is not available to use in the current study? If it was available, it would be valuable to analyse the association with aBMD of objectively measured physical activity in 2007-08 and 2015-16.

• Page 16, lines 297-300: The Sagelv et al. study referenced here showed only weak (albeit significant) correlations (r = 0.11 to 0.26). This should be acknowledged in this sentence.

• Table 1: Descriptive labels for the 4 physical activity categories would be more informative than level 1, level 2, level 3, level 4. E.g. as in Table 4 where they are labelled as inactive, light, moderate, vigorous activity.

• The footnote of Table 3 states Age groups are based on participants’ age at Tromsø6. A similar statement should also be included in the footnote for Table 2.

6. PLOS authors have the option to publish the peer review history of their article (what does this mean?). If published, this will include your full peer review and any attached files.

Reviewer #1: No

Reviewer #2: No

---

## [Author Response · Author response to Decision Letter 0]

18 Nov 2021

We thank the reviewers for very thorough and helpful comments. We have made our best efforts to address the comments and we believe that the revisions have improved the manuscript. Manuscript revisions are indicated by use of Track changes. Our responses to the reviewers are written in cursive under the respective comment and page numbers refer to the document with Track Changes.

Reviewer #1: Well written manuscript. Minor comment

The sample is probably a relatively healthier population than in Australia (or the USA) judged from their BMI below (in Australia 41% of subjects in this age group are obese based on BMI https://www.aihw.gov.au/reports/australias-health/overweight-and-obesity)

Study: Women had BMI of 26.7 ± 4.6 kg/m2 Men 27.1 ± 3.6 kg/m2

It is likely that the segment of this relatively non obese population that is most active is also the healthiest and would eat better, probably drink less, and possibly smoke less (although they tried to account for this). The authors are aware of this limitation and refer to it quicky in the final paragraph before the Conclusion when they say:

"Also, our analysis did not include additional confounders such as dietary factors, medication affecting bone metabolism or general health status."

This is a bit brief and should be expanded by saying it is possible that the more physically active cohort would most likely be the 'healthiest cohort' which is an alternative explanation for the findings.

Response: Thank you for taking the time to review our manuscript and for raising the issue of sample generalizability. We have now expanded this limitation about these issues regarding lifestyle and the more physically active cohort, page 16, line 307-311.

Reviewer #2: This is an interesting, rigorously conducted and well-reported study showing that higher physical activity levels are associated with greater hip aBMD in a large cohort of adults and older adults. The subjective categorisation of physical activity is a limitation of the study, however the authors have thoroughly discussed this in the limitations section. Some general and specific comments are provided below.

Response: Thank you for reviewing our manuscript and providing valuable feedback! 

General comments:

• There are very few participants categorised as physical activity level 4. I wonder whether combining physical activity categories 3 and 4 (as was done in the Morseth et al 2010 paper; reference 18) would have any influence on the results.

Response: By combining activity level groups 3 and 4, the results of the linear mixed models analysis become very similar to the results of the original analysis. Combining the two groups results in loss of information since we are also interested in effects concerning the highest activity level group. Thus we would prefer to not combine the groups.

• It would be interesting to see if physical activity is similarly associated with femoral neck aBMD. Including femoral neck BMD data would strengthen the study.

Response: The results from a linear mixed model analysis on femoral neck BMD yield very similar results and conclusions about the association between physical activity and BMD compared with the situation where total hip aBMD was used as outcome variable. This is expected since the correlation between total hip aBMD and femoral neck aBMD is approximately 0.90 on all measurement occasions. 

As a result of the reviewer’s suggestion, the regions included in total hip aBMD are now specified in the methods section (page 7, line 137-138) and correlation between femoral neck BMD and total hip BMD is added to the results (page 10, line 203-205). We also added associations between physical activity and femoral neck aBMD to the abstract (page 2, line 36-37), methods (page 7, line 138-139 and page 8, line 160) and results (page 10, line 202-203). Due to the high correlation between total hip aBMD and femoral neck aBMD, we have chosen not to report the femoral neck BMD linear mixed model results in the manuscript. 

• Reporting BMD results as percentage differences/changes (as opposed to absolute values in mg/cm2) would be more easily interpreted by the reader.

Response: We agree with the reviewer and this is now added to the manuscript, table 3 (former table 2), page 12 and table 4 (former table 3), page 13.

Specific Comments:

• Page 2, line 28: Given it is a key component of the study, I think it should be made clear in the abstract that the questionnaire categorised participants into physical activity levels (as opposed to estimated volume/intensity of physical activity). Wording such as ‘Categories of physical activity were determined by questionnaire’ would be more informative.

Response: We have now clarified the sentences, page 2, line 24-26. 

• The results section should be presented in the same order as the study aims. To align with the order of the aims presented at the end of the introduction, the ‘Changes in left total hip aBMD’ and ‘Physical activity and left total hip aBMD’ sections in the results should be swapped.

Response: This is a good suggestion that makes the manuscript more stringent. This is now swapped in the manuscript. Table 2 is currently table 3, table 3 is currently 4 and table 4 is currently 2. 

• Page 16, lines 296-300: It is mentioned here that physical activity was assessed by accelerometry in a small sample in 2007-08, and in a larger sample in 2015-16. The next sentence says that objective physical activity data was not available in 2007-08. Does this mean that the data used in the Emaus et al. paper cited is not available to use in the current study? If it was available, it would be valuable to analyse the association with aBMD of objectively measured physical activity in 2007-08 and 2015-16.

Response: In the Tromsø Activity sub-study, a sample of 313 men and women aged 40–44 years were randomly selected from Tromsø6 to validate the SGPALS against accelerometer. Unfortunately, only 39 participants had valid accelerometer data in both Tromsø6 and Tromsø7. Therefore, we chose not to use the accelerometer data from Tromsø6. 

• Page 16, lines 297-300: The Sagelv et al. study referenced here showed only weak (albeit significant) correlations (r = 0.11 to 0.26). This should be acknowledged in this sentence.

Response: This is added in the manuscript now, page 16, line 302-303. 

• Table 1: Descriptive labels for the 4 physical activity categories would be more informative than level 1, level 2, level 3, level 4. E.g. as in Table 4 where they are labelled as inactive, light, moderate, vigorous activity.

Response: Physical activity categories are now marked in Table 1, page 9-10 as in Table 2 (former Table 4), page 11. 

• The footnote of Table 3 states Age groups are based on participants’ age at Tromsø6. A similar statement should also be included in the footnote for Table 2.

Response: This is now added to former Table 2, current Table 3. Age groups are based on baseline, therefore following footnote for the current Table 3: “Age groups are based on participants’ age at T5”, page 12, line 227.

---

## [Decision Letter · Decision Letter 1]

20 Dec 2021

A 15-year follow-up study of hip bone mineral density and associations with leisure time physical activity. The Tromsø Study 2001-2016.

PONE-D-21-28316R1

Dear Dr. Mikkilä,

We’re pleased to inform you that your manuscript has been judged scientifically suitable for publication and will be formally accepted for publication once it meets all outstanding technical requirements.

Kind regards,

Robert Daniel Blank, MD, PhD

Academic Editor

PLOS ONE

Additional Editor Comments (optional):

Reviewers' comments:

Reviewer's Responses to Questions

**Comments to the Author**

1. If the authors have adequately addressed your comments raised in a previous round of review and you feel that this manuscript is now acceptable for publication, you may indicate that here to bypass the “Comments to the Author” section, enter your conflict of interest statement in the “Confidential to Editor” section, and submit your "Accept" recommendation.

Reviewer #1: All comments have been addressed

Reviewer #2: All comments have been addressed

2. Is the manuscript technically sound, and do the data support the conclusions?

Reviewer #1: Yes

Reviewer #2: Yes

3. Has the statistical analysis been performed appropriately and rigorously? 

Reviewer #1: Yes

Reviewer #2: Yes

4. Have the authors made all data underlying the findings in their manuscript fully available?

Reviewer #1: No

Reviewer #2: No

5. Is the manuscript presented in an intelligible fashion and written in standard English?

Reviewer #1: Yes

Reviewer #2: Yes

6. Review Comments to the Author

Reviewer #1: All comments have been addressed satisfactorily. Well written manuscript and is ready for publication.

Reviewer #2: (No Response)

7. PLOS authors have the option to publish the peer review history of their article (what does this mean?). If published, this will include your full peer review and any attached files.

Reviewer #1: No

Reviewer #2: No

---

## [Editor Report · Acceptance letter]

26 Dec 2021

PONE-D-21-28316R1 

A 15-year follow-up study of hip bone mineral density and associations with leisure time physical activity. The Tromsø Study 2001-2016. 

Dear Dr. Mikkilä:

I'm pleased to inform you that your manuscript has been deemed suitable for publication in PLOS ONE. Congratulations! Your manuscript is now with our production department. 

Kind regards, 

on behalf of

Professor Robert Daniel Blank 

Academic Editor

PLOS ONE